# An Improved Mobile Mapping System to Detect Road-Killed Amphibians and Small Birds

**Diana Sousa Guedes \*** , **Hélder Ribeiro and Neftalí Sillero**

Centro de Investigação em Ciências Geo-Espaciais (CICGE), Faculdade de Ciências da Universidade do Porto, 4430-146 Vila Nova de Gaia, Portugal; helder.ribeiro@fc.up.pt (H.R.); neftali.sillero@gmail.com (N.S.)
\* Correspondence: dianasguedes@gmail.com

**Abstract:** Roads represent a major source of mortality for many species. To mitigate road mortality, it is essential to know where collisions with vehicles are happening and which species and populations are most affected. For this, moving platforms such as mobile mapping systems (MMS) can be used to automatically detect road-killed animals on the road surface. We recently developed an MMS to detect road-killed amphibians, composed of a scanning system on a trailer. We present here a smaller and improved version of this system (MMS2) for detecting road-killed amphibians and small birds. It is composed of a stereo multi-spectral and high definition camera (ZED), a high-power processing laptop, a global positioning system (GPS) device, a support device, and a lighter charger. The MMS2 can be easily attached to any vehicle and the surveys can be performed by any person with or without sampling skills. To evaluate the system's effectiveness, we performed several controlled and real surveys in the Évora district (Portugal). In real surveys, the system detected approximately 78% of the amphibians and birds present on surveyed roads (overlooking 22%) and generated approximately 17% of false positives. Our system can improve the implementation of conservation measures, saving time for researchers and transportation planning professionals.

**Keywords:** road ecology; robotics; computer vision; conservation; mitigation; wildlife-vehicle collision

## 1. Introduction

Roads negatively affect wildlife, from direct mortality to habitat fragmentation [1,2]. Mortality caused by collision with vehicles on roads is a major threat to many species [3]. Some groups are particularly vulnerable like amphibians and birds due to their ecological characteristics and foraging/hunting habits [4–7].

Monitoring wildlife-vehicle collisions (WVCs) is essential to establish correct mitigation measures. Many countries have national monitoring systems for identifying species' mortality hotspots [8,9]. These surveys are performed by car at low speed (20–30 km/h) and are therefore expensive and time-consuming. The need to stop at every detected WVC makes the process somewhat dangerous for the surveyors, mainly during the survey of amphibians on rainy nights (e.g., [6,7]). Furthermore, identification errors may occur because the users' skills to detect and identify the animals are not equal [10]. Moreover, many WVCs, namely the smallest, are very difficult to detect and are easily overlooked. Therefore, more efficient methods for surveying roads over larger areas and for reducing identification errors are necessary.

For broader and continuous monitoring programs, surveys should be logistically and economically achievable. Robotics and computer vision can provide the necessary tools to automatically detect wildlife-vehicle collisions by passive surveys, with the consequent reduction in costs and resources [11,12]. Moving platforms such as mobile mapping systems (MMSs) can take images from

the road surfaces, automatically identify objects using intelligent algorithms, and extract the necessary geospatial data to locate WVCs [13,14].

We recently developed a MMS to detect wildlife-vehicle collisions of amphibians, composed of a scanning system on a trailer [11,12]. The system included a trichromatic line scan camera, a light-emitting diode, a global positioning system (GPS) device, and an industrial computer. This system created a continuous road surface image, obtaining an object resolution of less than one millimeter [11]. Real surveys provided a correct classification rate of amphibians of 84% and a rate of failed classification of 16% [12]. One main advantage of this system is the possibility of checking the recorded images to find overlooked animals back at the office, something impossible to do in traditional surveys [6,7]. This system is ideal for passive surveys, which are more economical and less time-consuming [12]. Anyone with or without sampling skills can survey while travelling. However, the system has some limitations: the size and weight makes it cumbersome, the road sampling width is not large (the camera only captures one meter of the road width, not enough to cover an entire one-way road), the sampling length is limited to an external power source capacity (approximately one hour), and the total price of the system may be somewhat expensive (15,000 €) [11,12].

We present here an improved version of a mobile mapping system (MMS2) for detecting amphibians and small birds. This new version has a considerably reduced size and weight (see Figure 1). Aside from the size, the MMS2 has other improvements to the previous version: it can be easily attached to the back of any vehicle, it has a more straightforward workflow and unlimited energetic consumption, the sampling width effectively covers a one-way national road, and the system is considerably cheaper (2000 €), making it easier to implement elsewhere. Furthermore, we have improved the detection algorithm of animals on roads and also increased the image database. Due to the unlimited sampling length and less time-consuming survey process, the system can be used to cover wider areas (e.g., placing the MMS2 in commercial vehicles with long daily travels). While humans may be able to detect and identify an animal more accurately, the verification of thousands of images is time-consuming and hampers the research. For this, computer vision algorithms with automatic animal detection have an essential role.

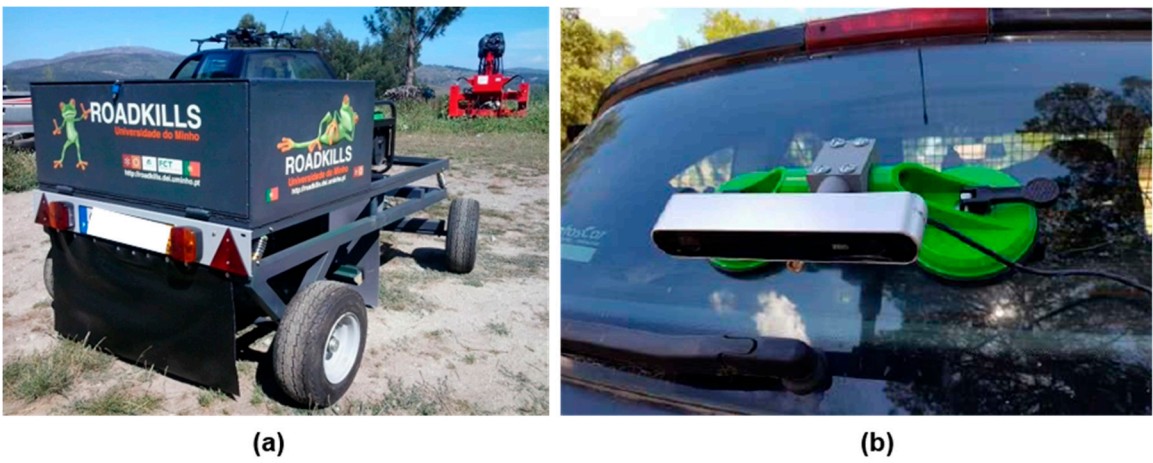

**Figure 1.** Image of: (**a**) the mobile mapping system 1 (MMS1) outdoors and (**b**) the mobile mapping system 2 (MMS2) attached to the back of a vehicle. The complete MMS2 system also includes a laptop and a lighter charger that go inside the vehicle.

## 2. Materials and Methods

### 2.1. Description of the Mobile Mapping System 2

The mobile mapping system 2 (MMS2) was designed to automatically detect wildlife-vehicle collisions of amphibians and small birds on roads (Figure 1b). It includes:

(1) A ZED stereo camera (Stereolabs Inc): a lightweight (159 g), dual 4 MP camera with a maximum high-resolution of 4416 × 1242 pixels at a maximum frequency of 100 Hz. This camera has a wide-angle video and depth view (110°), a depth perception up to 20 mm, and 6-DoF positional tracking. It captures at a video resolution of 1080 p HD at 30 FPS or 2 K at 15 FPS. The sensor format is 16:9 for a horizontal field of view. The battery consumption is 5 V/380 mA;

(2) High-power processing computer (GL553VD ASUS laptop): laptop with a GTX1050 GPU from Nvidia, and an Intel I7-7700HQ with 8 CPUs of 2.8 GHz with 16 GB of RAM;

(3) GPS receptor: GPS SIM808 (SIMCOM) and Glonass receptor with location-based service (LBS) positioning and omni-positioning. It retrieves the GPS coordinates of the MMS2;

(4) Attachment device: sucker support for bicycles; and

(5) Lighter charger for vehicles for unlimited sampling length (does not depend on an external power source, but rather depends on the car battery; the system runs uninterruptedly as long as the vehicle is working).

A diagram of the system architecture is presented in Figure 2. The system is controlled by several applications: mobile software to turn the device off and on, desktop software to collect and save images, desktop software to detect the WVCs in images, and software that continuously pin-points the device coordinates along the way.

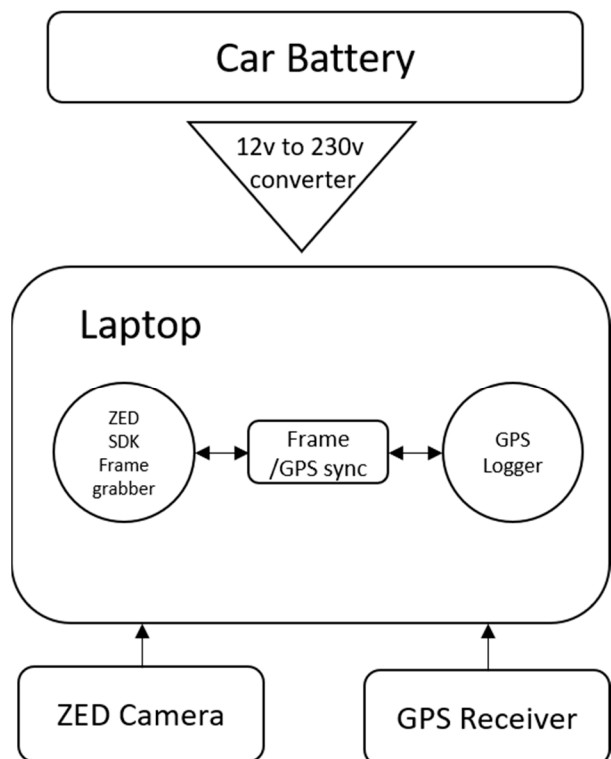

**Figure 2.** Diagram representing the system architecture of the MMS2.

### 2.2. Algorithms

We used the state-of-the-art machine learning computer vision algorithm convolutional neural network (CNN, a class of deep neural networks; [15]) to automatically detect WVCs on roads. CNNs are regularized versions of multilayer perceptrons (i.e., fully connected networks where each layer of neurons is connected to all neurons in the next layer; [15,16]). This self-learning algorithm needs to be trained with images of live animals, road-killed animals, and any object likely to be found on roads (e.g., garbage thrown away by drivers). This architecture is capable of handling small sets of images

(commonly between 1000 and 3000 images). However, with a greater image database, the algorithm efficiency usually increases.

We performed some trials with several state-of-the-art deep learning image classifiers: VGG16, VGG19 (Visual Geometry Group; [17]), ResNet50 (Residual Networks; [18]), Inception V3 [19], and Xception [20]. For that, we used 150 images, 25 of which contained animals. Then, we measured the efficiency of each image classifier in detecting the animals with the Keras library pre-trained models [21]: the VGG16 detected approximately 62% of the animals, the VGG19 42%, the ResNet 35%, the Inception V3 62%, and the Xception 23%. We thus selected the VGG16, which showed the best efficiency (parameters: weights, 0.0005; input tensor, true; pooling, max). The VGG16 is a simple network, using only $3 \times 3$ convolutional layers stacked on top of each other in increasing depth [17].

We trained the algorithm with 1296 collected images of road-killed and alive amphibians and birds obtained from the University of Évora image database and from the RoadKills project database. In addition, we also fed the algorithm with 1250 recorded images by the MMS2. In total, we trained the algorithm with 2546 images of road-killed and live amphibians and birds.

We used the max pooling process to reduce the pixel density of the images, thereby reducing the computing time and minimizing model overfitting [22]. Max pooling is a sample-based discretization process with the aim of down-sampling an image, thus reducing its spatial size (i.e., width and height). It is common to insert pooling layers between consecutive convolutional layers in a CNN architecture [22]. Convolutional layers are the main building blocks of a convolutional network, where each neuron is connected to local neurons in the previous layer and the same set of weights is used for every neuron. Two fully connected layers (i.e., layers where each neuron is connected to every neuron in the previous layer) are then followed by a SoftMax classifier (a generalization of the binary form of logistic regression that delivers the probabilities for each class; a higher probability means a higher confidence that the image belongs to that class; [23,24]). We also applied batch normalization and dropout. Batch normalization normalizes the activations of a given input image before passing it to the next layer in the network. It is very effective at reducing the number of epochs (number of times the algorithm sees the entire dataset) required to train a CNN as well as stabilizing training itself. Our model ran on 25 epochs with the 2546 images of the training database. Dropout forces the network to become more robust. It is the process of disconnecting random neurons between layers. This reduces overfitting (i.e., reduces the number of false positives), increases the accuracy of animal detection (i.e., increases the rate of correct classification and decreases the rate of failed classification), and allows our network to generalize better for unfamiliar images.

*2.3. Testing Framework*

Some preliminary tests were performed in order to find the optimal luminosity and color configuration of the camera. For that, we went through the surveyed roads on different days with different light and weather conditions. Then, we verified the captured images, changing the camera configurations every time (exposure, white balance, brightness, contrast, hue, and saturation), until we found the optimal combination of parameters. After that, we tested the efficiency of the MMS2 in detecting small birds and amphibians using controlled and real surveys.

For the controlled situations, we used dead specimens from the collection of the University of Évora: 10 specimens of amphibians, 10 of birds, five of mammals, five of reptiles, and 10 items of garbage frequently found on roads. We performed 12 surveys in a small road section in Évora (South Portugal; Figure 3), with different vehicle speeds (from 20 to 100 km/h) in order to find the most appropriate settings (Table 1). Additionally, we randomly distributed 58 dead specimens of eight bird species (*Carduelis carduelis*, *Cyanistes caeruleus*, *Erithacus rubecula*, *Passer domesticus*, *Sylvia atricapilla*, *Sylvia melanocephala*, *Strix aluco*, and *Tyto alba*) on two paved roads and its road verges (roads N4 and M529; Figure 3). These species were selected because they are the most commonly found road-kill in the Évora region (António Mira, personal communication). We schematically noted the location of each specimen for further comparison with the recorded images.

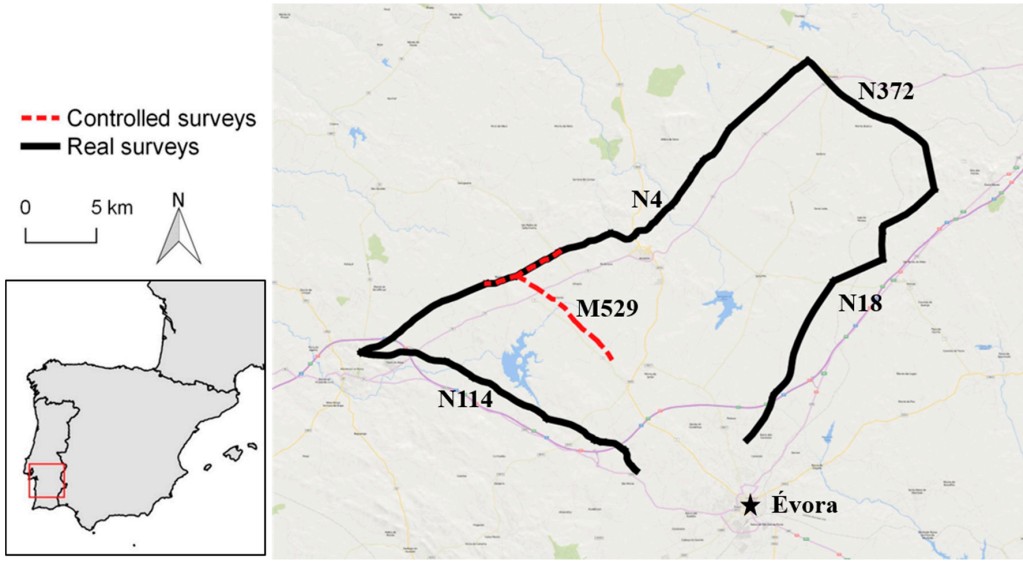

**Figure 3.** Study area and surveyed roads in controlled and real surveys.

**Table 1.** Camera configuration and results of the controlled surveys. The best configuration resulted in a correct classification rate of 80% and originated 20% of false positives (in bold). Resolution: camera resolution; FPS: frames per second; Speed: approximate vehicle speed; Total images: total number of frames produced; Images with animals: total number of frames with the presence of wildlife-vehicle collisions (WVCs); True positives: number of frames the algorithm detected a WVC correctly; Failed classification: number of frames the algorithm overlooked a WVC; False positives: number of frames in which the algorithm detected a WVC incorrectly; Correct classification: percentage of true positives from the total number of images with animals.

| Resolution | FPS | Speed | Total Images | Images with Animals | True Positives | Failed Classification | False Positives | Correct Classification |
|---|---|---|---|---|---|---|---|---|
| 2208 × 1242 | 60 | 80 km/h | 1406 | 53 | 39 | 14 (26.4%) | 9 (18.8%) | 73.5% |
| **2208 × 1242** | **30** | **60 km/h** | **980** | **25** | **20** | **5 (20%)** | **5 (20%)** | **80%** |
| 1920 × 1080 | 60 | 30 km/h | 1250 | 42 | 31 | 11 (26.2%) | 7 (18.4%) | 73.8% |
| 1920 × 1080 | 30 | 60 km/h | 680 | 35 | 27 | 8 (22.9%) | 6 (18.2%) | 77.1% |
| 1280 × 720 | 60 | 70 km/h | 1560 | 32 | 24 | 8 (25%) | 6 (20%) | 75% |
| 1280 × 720 | 30 | 30 km/h | 850 | 42 | 31 | 11 (26.2%) | 7 (18.4%) | 73.8% |
| 1280 × 720 | 80 | 20 km/h | 2230 | 52 | 37 | 15 (28.8%) | 9 (19.6%) | 71.2% |
| 1280 × 720 | 60 | 100 km/h | 1120 | 30 | 19 | 11 (36.7%) | 6 (24%) | 63.3% |
| 2208 × 1242 | 60 | 30 km/h | 1420 | 25 | 19 | 6 (24%) | 4 (17.4%) | 76% |
| 1920 × 1080 | 60 | 30 km/h | 1600 | 26 | 19 | 7 (26.9%) | 5 (20.8%) | 73.1% |
| 2208 × 1242 | 30 | 40 km/h | 820 | 32 | 25 | 7 (21.9%) | 6 (19.4%) | 78.1% |
| 1920 × 1080 | 30 | 40 km/h | 905 | 34 | 26 | 8 (23.5%) | 6 (18.8%) | 76.5% |

Finally, we performed several real surveys on national roads in Évora (Figure 3) with a total distance of 720 km for 12 days. The real surveys were always performed at a vehicle speed between 40 and 80 km/h. After our vehicle with the MMS2 travelled through the selected roads, a vehicle from the University of Évora wildlife-vehicle collisions monitoring program (LIFE LINES project [25]) repeated the same path to cross information.

In all situations, we manually checked the images with WVCs (in real surveys, using the GPS coordinates collected by the University of Évora whenever they detected a WVC) and the ones the algorithm classified as WVCs. In each trial, we manually counted the number of frames with WVCs.

## 3. Results

After some preliminary tests, we selected the optimal configuration for the camera luminosity and color (automatic adjustment of exposure, white balance and gain, brightness = 1, contrast = 6, hue = 0, saturation = 2). The best configuration for a person to identify correctly the animal, in relation to

image quality and sharpness, was a camera resolution of 2208 × 1242 and a vehicle speed of around 60 km/h. This configuration produced images with a pixel size of 0.002 mm, and a horizontal field of view of 76°. The MMS2 generated an image at every 60 cm of a given road, resulting in at least three images for each object on the road.

The controlled surveys with different combinations of camera resolution and vehicle speed resulted in a correct classification rate (i.e., percentage of detected animals) between 63.3 and 80% and a percentage of false positives (i.e., percentage of frames where the algorithm detected a WVC incorrectly) between 17.4 and 24% (Table 1). The algorithm excluded between 20 and 36.7% of images with WVCs (rate of failed classification, i.e., percentage of overlooked WVCs). The configuration with the best performance was the combination of a resolution of 2208 × 1242, 30 FPS, and a vehicle speed of 60 km/h with 80% of correct classifications, 20% of failed classifications, and 20% of false positives (Table 1).

The real surveys resulted in a rate of correct classification of 78%, overlooking therefore 22% of the animals on surveyed roads (Figure 4). Approximately 17% of the images classified with WVCs were false positives. The algorithm classified 37.1% of the detected animals to the species level (the remainder was classified as "others"). Together, the controlled surveys and real surveys resulted in more than a million road images. We compiled the main differences between the MMS1 and the MMS2 in Table 2.

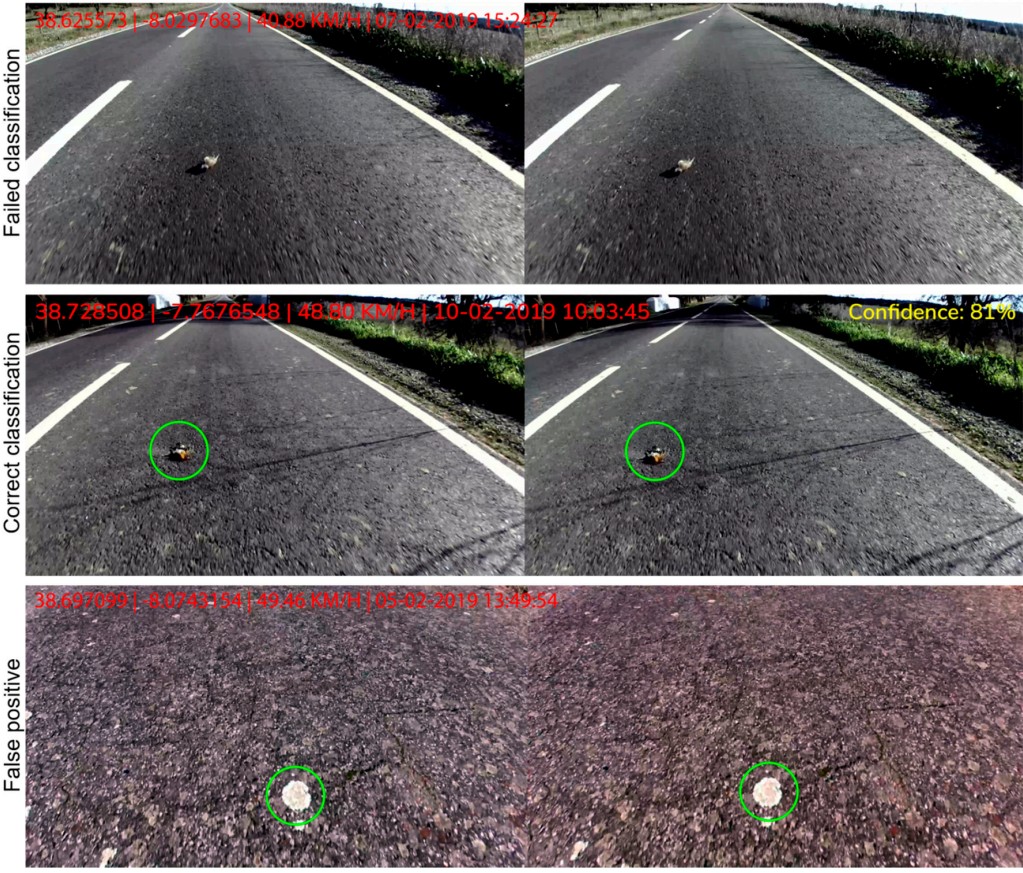

**Figure 4.** Examples of images captured by the MMS2 during real surveys: in the two upper images, the algorithm failed to detect the WVC (failed classification); in the middle images, the algorithm correctly detected the WVC (correct classification); and in the lower ones, the algorithm detected an animal incorrectly (false positive). The ZED camera is a dual-camera, always providing two images with slightly different angles.

**Table 2.** Comparison and main differences between the MMS1 and the MMS2 [11,12].

|  | **MMS1** | **MMS2** |
|---|---|---|
| Size | See Figure 1a | See Figure 1b |
| Approximate cost | 15.000 € | 2.000 € |
| Sampling length | Approximately 1 h/30 km | Unlimited |
| Sampling width | 1 m | 6–7 m |
| Maximum vehicle speed | 70 km/h | 60 km/h |
| Adequate to all weather and light conditions | Yes | No |
| Attachable to any car | No | Yes |
| Mean rate of correct classification on real surveys | 84% | 78% |
| Mean percentage of false positives on real surveys | 80% | 17% |

## 4. Discussion

Machine learning and computer vision are increasingly being applied in ecology for conservation purposes (e.g., [24,26]). For instance, two common applications are camera traps with the purpose of studying species' ecology or the verification of uploaded images by citizen scientists, with automatic detection and identification of the species [24,26]. Most algorithms that have been developed to identify animals are trained to detect alive animals [26], which is easier because individuals of the same species rarely have different aspects. Both for humans and algorithms, detecting and identifying WVCs is much more complex and challenging as in many cases, the specimens are in very bad condition.

This improved version of the mobile mapping system has great potential for national monitoring surveys in search of WVCs. The algorithm has good effectiveness in detecting birds and amphibians with very high animal detectability. In controlled surveys, with the best system configuration, the MMS2 detected approximately 80% of the amphibians and small birds on surveyed roads (Table 1), while in real surveys, the system detected on average 78% of the animals. The MMS2 overlooked around 22% of animals in real surveys, which was worse than the previous version (the MMS1 overlooked 16% of present amphibians; [12]). The MMS1 has the great advantage of having a lighting system, equally illuminating every part of the image. This, together with the better image resolution and lower image distortion, may be the reason for a higher rate of correct classification with the MMS1. However, 22% of overlooked animals is still an acceptable error as in traditional surveys, it is never really known how many animals are missed. Furthermore, despite being a time-consuming process, the system allows for the checking of all images after the survey, or at least the true and false positives. Similar to the MMS1, this system worked properly at 60 km/h (the MMS1 worked well at 70 km/h as maximum). At higher speeds, the image quality, sharpness, and focus might be compromised, hampering the correct identification of the road-killed animal.

The biggest improvement of the MMS2 over the previous version (besides the reduction in size and cost and the increase in the sampling width; see Table 2) is the great reduction in false positives. The MMS1 created around 80% of false positives [12], while the MMS2 produced approximately 17% of false positives. This is a great advantage, as the manual verification of all false positives is very time-consuming. The MMS1 created images with better resolution (road surface resolution under a millimeter [11]) in comparison with the images obtained with the MMS2. Furthermore, the image resolution by MMS2 depends on how the device is attached to the car, while the position of the MMS1 in the trailer was fixed. Nevertheless, the number of false positives in the MMS2 was greatly reduced, possibly due to the improvement of the detection algorithm and to the bigger image database for training the algorithm. Moreover, the algorithm of the MMS2 is much more robust than the previous version, which only uses simpler image processing and computer vision methods and not machine learning.

However, there are some disadvantages of this system: (1) as in the previous version, it still requires human intervention to determine if the animal is alive or road-killed; (2) the weather conditions may affect the images' quality, hampering the correct animal identification: harsh light hampers the

camera to rapidly adjust the exposure, while very humid conditions may accumulate water drops in the camera, with the need to clean it once in a while; (3) the system is not suitable to use at night: there is no light system as in the previous version; (4) a high storage capacity is necessary (the tests originated 1.65 terabytes of images); (5) specialized people are needed for image processing and animal identification; and (6) a workstation is necessary for computing heavy data. Nevertheless, with the growth of the image database, the algorithm will continue to improve, with a continuous reduction in errors. The new algorithm has been trained to specifically detect road-killed amphibians and birds on roads, but as in the previous version [12], it could potentially be trained to detect any animal group, object, or road feature.

The implementation of an easy to use system and a straightforward methodology for monitoring wildlife-vehicle collisions is needed to better apply and monitor the success of mitigation measures [27]. Our system can be used on national roads everywhere and may contribute to the improvement of national wildlife-vehicle collision databases that can be used for determining priority sites, and species or populations for conservation. Moreover, our system can save substantial amounts of time for researchers and transportation planning professionals. We hope that our system will be used to increase the data and research on road ecology.

**Author Contributions:** Conceptualization, Hélder Ribeiro and Neftalí Sillero; Data curation, Diana Sousa Guedes and Hélder Ribeiro; Formal analysis, Hélder Ribeiro; Methodology, Diana Sousa Guedes, Hélder Ribeiro, and Neftalí Sillero; Project administration, Neftalí Sillero; Software, Hélder Ribeiro; Supervision, Neftalí Sillero; Writing—original draft, Diana Sousa Guedes; Writing—review & editing, Diana Sousa Guedes, Hélder Ribeiro, and Neftalí Sillero.

**Funding:** This work was funded by the LIFE LINES project (LIFE 14/NAT/PT/001081). DSG was supported by research grants from LIFE LINES and ENGAGE-SKA (POCI-FEDER 022217). HR was supported by a research grant from LIFE LINES and a contract by ENGAGE-SKA. NS is supported by a a contract from Fundação para a Ciência e Tecnologia (CEECIND/02213/2017).

**Acknowledgments:** We thank António Mira and all the LIFE LINES and MOVE team from the University of Évora for their helpful contribution. We also thank the ROADKILLS project team (PTDC/BIA-BIC/4296/2012). We thank James Harris for checking the English in the manuscript.

**Conflicts of Interest:** The authors declare no conflicts of interest.

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
