# Peer review of "An Improved Mobile Mapping System to Detect Road-Killed Amphibians and Small Birds"

_ijgi, doi:10.3390/ijgi8120565_

Round 1

Reviewer 1 Report

Thanks to the editors for inviting me to review this interesting manuscript. The concept of the manuscript is quite useful for a practical purpose. However, I think the manuscript still needs a bit more details to properly present its work. Some key issues:

The introduction needs to discuss the importance of developing this system. It pointed out the cost, but I am wondering it underrepresents its value. For example, the new device can be used for wider coverage of an area. I think their previous paper in Sensor,  Lopes et al., (2016) did a better job. Also, the authors need to discuss the existing body of work related to this new sensor. For example, what other areas use this kind of method? Are any video-based systems used to detect animals?  Description of MMS2 is too brief. Maybe a figure and an image of the system would help to visualise the system. Also how they are connected where the processing happens are unclear. I would recommend adding a figure like the Lopes et al., (2016) paper.   I have a serious concern about the algorithm. First, the authors did not provide any details about the justification for classifier selection, are all of them softmax, if yes, why, did the authors tried others, such as RELU, Gaussian? Second, the model running parameters are bit unclear, which software system used? Are they built upon the pre-trained models? How many epochs the model run? I just saw details about the training set, was the model has any test set? What are the accuracies of training and tests? Also, considering the deep learning network structure, the training set seems quite small, how the authors deal with smaller set for a large network? I think the authors need to provide these details to ensure their model is robust.  The result is too brief! The result section should give some details of the experimental set-up. Few images collected by the MMS2, and maybe some example wich images came false positive.  Discussion can be further improved by discussing why the false positive may increase for this system than MMS1? Some indication of how much time this system can save?  How to improve the model (e.g. change structure, increase training set).

Overall I think the MMS2 system has improved some of the aspects of MMS1, it is cost-effective. But to achieve its full potential, the authors need to make the process more transparent, which can help replication of the system. Additionally, the issues pointed above should be considered to increase the readability of the paper to experts from multiple disciplines. 

Author Response

Response to Reviewer 1 Comments

Point 1: Thanks to the editors for inviting me to review this interesting manuscript. The concept of the manuscript is quite useful for a practical purpose. However, I think the manuscript still needs a bit more details to properly present its work. Some key issues: The introduction needs to discuss the importance of developing this system. It pointed out the cost, but I am wondering it underrepresents its value. For example, the new device can be used for wider coverage of an area. I think their previous paper in Sensor,  Lopes et al., (2016) did a better job.

Authors: Thank you very much for accepting to review our manuscript and for your comments. We have highlighted in yellow every modification in the manuscript. We have now clarified the importance and value of the system in the introduction: in the last paragraph of the introduction, line 4 (“it has a more straightforward workflow”), line 5-6 (“making it easier to implement elsewhere”), lines 7-11 (“Due to the unlimited sampling length and less time-consuming survey process, the system can be used to cover wider areas (e.g. placing the MMS2 in commercial vehicles with long daily travels). While a human may be able to detect and identify an animal more accurately, the verification of thousands of images is time-consuming and hampers the research. For this, computer vision algorithms with automatic animal detection have an essential role.”)

Point 2: Also, the authors need to discuss the existing body of work related to this new sensor. For example, what other areas use this kind of method? Are any video-based systems used to detect animals? 

Authors: We have now discussed the applicability of machine learning and image-based systems to detect animals in the discussion (first paragraph).

Point 3: Description of MMS2 is too brief. Maybe a figure and an image of the system would help to visualise the system.

Authors: We have now added a figure of the system attached to a car (Figure 1b).

Point 4: Also how they are connected where the processing happens are unclear. I would recommend adding a figure like the Lopes et al., (2016) paper.  

Authors: We have now added a figure illustrating the system architecture (Figure 2).

Point 5: I have a serious concern about the algorithm. First, the authors did not provide any details about the justification for classifier selection, are all of them softmax, if yes, why, did the authors tried others, such as RELU, Gaussian?

Authors: We used Convolutional Neural Networks (CNN) because they are the most common used on object identification. A simple test was done using 150 images, from which 25 images contained animals. Then, we tried different CNN architectures, and measured the efficiency of these architectures in detecting the animals (VGG19: 42%, ResNet50: 35%, Inception V3: 62%, 97 Xception: 23%, VGG16: 62%). This was done with the Keras library pre-trained models, and using a set of 600 images to post train the algorithm. The VGG16 presented better results so we decided to follow its path. At this stage, the false positives where not taken in account as they could be improved later. We included this in the methods section of the manuscript (algorithms, second paragraph, lines 3-6).

Point 6: Second, the model running parameters are bit unclear, which software system used? Are they built upon the pre-trained models? How many epochs the model run?

Authors: The following model parameters used were: Weights: 0.0005; input tensor: true; pooling: MAX. We used the Keras library VGG16 model, with weights pre-trained on ImageNet challenge with 1000 classes already trained (some containing animals). The post-training was done using our subset of images (amphibians and birds). The model run on 25 epochs with the 2546 images of the training database. We have now explained this in the methods section (algorithm, second paragraph, lines 3-7 and fourth paragraph, lines 10-11).

Point 7: I just saw details about the training set, was the model has any test set?

Authors: As stated in the manuscript, the algorithm was trained with 2546 images. As the image database is not that large, we decided to use all the images for training the algorithm, not including any test set.  

Point 8: What are the accuracies of training and tests?

Authors: The accuracy of the training set of the controlled surveys was between 63.3 and 80% and of the real surveys was 78%. As stated before, we did not split the image database in training and test sets.

Point 9: Also, considering the deep learning network structure, the training set seems quite small, how the authors deal with smaller set for a large network? I think the authors need to provide these details to ensure their model is robust. 

Authors: The selected architecture was designed because it is capable of handling a small subset of images on the database. It is quite normal for a VGG16 to use a database between 2000 to 3000 images. We included this in the manuscript (methods, algorithms, first paragraph, line 6-8).

Point 10: The result is too brief! The result section should give some details of the experimental set-up. Few images collected by the MMS2, and maybe some example wich images came false positive. 

Authors: We have now added more details on the results section and some images collected by the MMS2 to the manuscript (Figure 4).

Point 11: Discussion can be further improved by discussing why the false positive may increase for this system than MMS1?

Authors: The possible reason for the reduction of the number of false positives is the improvement of the algorithm and the bigger image database. The algorithm is also much more robust than the previous version, which did not use artificial intelligence as the MMS2 do but rather used normal image processing and computer vision methods. We have now included this in the discussion (third paragraph, lines 5-11).

Point 12: Some indication of how much time this system can save?  How to improve the model (e.g. change structure, increase training set).

Authors: Considering the route of the road-kill monitoring program of University of Évora, this system would take around 2 hours of survey per day and around 7/8 hours of image processing. In comparison with regular surveys that takes on average 6 hours a day, this system would save around 4 hours of daily survey. Road-kill surveying is exhausting and can lead to several identification errors. Image processing in a workstation, once mechanized, would not be so tiring and would not be necessary to do it every day. The system would be improved with the increase of the image database. We included this in the discussion (fourth paragraph, line 8-9): “with the growth of the image database, the algorithm will continue to improve, with continuous reduction of errors.”

Point 13: Overall I think the MMS2 system has improved some of the aspects of MMS1, it is cost-effective. But to achieve its full potential, the authors need to make the process more transparent, which can help replication of the system. Additionally, the issues pointed above should be considered to increase the readability of the paper to experts from multiple disciplines.

Authors: We appreciate the referees’ comments and have now made the suggested improvements to the manuscript.

Reviewer 2 Report

This paper shows an improved version of a road-kill detection system. Results are significative, showing that the system performs its purpose. Also, the economic analysis is valuable for future reference. Nevertheless, as much as I consider respectable and useful any practical work, the paper does not convey this work. There are no figures in the paper, missing many great opportunities both for showing off the great development work and explaining better the concepts. This paper, being about image processing, should be full of images. First of all, I am missing a figure that represents the overall system architecture. This is basic in any paper showing a complex system, since it will help readers to understand each part independently. Another figure should show how pictures are taken and how they will feed the classifier. Apart from these two figures, there should be examples of false positives, false negatives and correct detections in the results section. In line 111, the training set is briefly described. This description should be much more comprehensive, including the distribution of different types of image (dead/alive, species, etc.), and, of course, a figure would help. Another picture or map of the location where the tests were performed would be informative. The text from line 171 until 182 is misplaced, it should be either in the introduction or on Section 2.1. Again, a side-by side image comparing MMS and MMS2 would be very helpful.

Although the paper contains enough details on the system, a correct use of figures would make it much more readable. Adding them and ordering a bit the ideas in the paper will help a lot in the next round.

Author Response

Response to Reviewer 2 Comments

Point 1: This paper shows an improved version of a road-kill detection system. Results are significative, showing that the system performs its purpose. Also, the economic analysis is valuable for future reference. Nevertheless, as much as I consider respectable and useful any practical work, the paper does not convey this work. There are no figures in the paper, missing many great opportunities both for showing off the great development work and explaining better the concepts. This paper, being about image processing, should be full of images. First of all, I am missing a figure that represents the overall system architecture. This is basic in any paper showing a complex system, since it will help readers to understand each part independently.

Authors: Thank you very much for accepting to review our manuscript and for the useful suggestions. We have highlighted in yellow every modification in the manuscript. We have now included several figures in the manuscript, including one of the system architecture (Figure 2).

Point 2: Another figure should show how pictures are taken and how they will feed the classifier. Apart from these two figures, there should be examples of false positives, false negatives and correct detections in the results section.

Authors: We have now included three examples of images captured by the MMS2: one with a failed classification, a second one with a correct classification and a third one with a false positive (Figure 4).

Point 3: In line 111, the training set is briefly described. This description should be much more comprehensive, including the distribution of different types of image (dead/alive, species, etc.), and, of course, a figure would help.

Authors: We have now indicated that the images were obtained from the University of Évora database and from the Roadkills project. Because these databases are not public, we cannot include examples of images used to train the algorithm, but we did include images of road-killed animals captured by the MMS2 (Figure 4). Although we agree with the reviewer, we decided to not discriminate the training set because we do not have this complete information yet, and it will take several days to analyse each image manually in order to determine the number of alive/dead animals and species.

Point 4: Another picture or map of the location where the tests were performed would be informative.

Authors: We have now included in the manuscript a figure of the study area with the surveyed roads (Figure 3).

Point 5: The text from line 171 until 182 is misplaced, it should be either in the introduction or on Section 2.1.

Authors: We have now deleted most of the paragraph from line 171 until 182. We left in the discussion (third paragraph) the reference to the reduction of false positives, as it would not make sense to place it in the introduction/methods.

Point 6: Again, a side-by side image comparing MMS and MMS2 would be very helpful.

Authors: We have now included an image comparing the MMS1 and MMS2 (Figure 1) and a table with the main differences between both systems (Table 2).

Point 7: Although the paper contains enough details on the system, a correct use of figures would make it much more readable. Adding them and ordering a bit the ideas in the paper will help a lot in the next round.

Authors: Thank you very much for the comments and suggestions. We have now introduced the suggested figures and reorganized the manuscript.

Reviewer 3 Report

The manuscript describes a new platform for automated detection of dead amphibians and small birds along roads as  a results of car collision. See the attached file for additional comments

My major remarks are:

Novelty - the major novelty arouse from the text is the technological improvement while giving little attention to the machine learning algorithm and process, in particular in the discussion section. There is an assumed novel methodology but it should be further stressed and discussed.

In some cases values and figures are missing while in other cases, they are not interpreted clearly. I assume there are important results but when statements are not supported by findings it makes them pretty weak. on the other hand - a statement like that in line 192 ('it can be trained to detect any animal group, objects or road features') is risky to be made if you have not tested that yourself or can rely on previous work.

What is the applicability of the system and technique to other parts off Evora? A short discussion is missing 

Throughout the manuscript, the authors give provide poor explanations concerning the results. They take it for granted that the reader would understand thus shortening descriptions. They should elaborate more (see the attached file for details)

I am missing a Map of the area (Evora) including N4 and M529 and the rest of the area where the tests were implemented

Table 1 contains the results of the controlled tests. Perhaps it would be beneficial to include also the results of the real detection in a summarized table as well

English and grammar - should be improved. I have made several corrections but language editing is essential 

Author Response

Response to Reviewer 3 Comments

Point 1: The manuscript describes a new platform for automated detection of dead amphibians and small birds along roads as a results of car collision. See the attached file for additional comments. My major remarks are: Novelty - the major novelty arouse from the text is the technological improvement while giving little attention to the machine learning algorithm and process, in particular in the discussion section. There is an assumed novel methodology but it should be further stressed and discussed.

Authors: We appreciate the referee’s constructive comments. We have highlighted in yellow every modification in the manuscript. We have now introduced in the first paragraph of the discussion the importance of applying computer vision in detecting animals. We have explained briefly how the algorithm works and what parameters were used but without entering in an exhausting description as the target readers would be conservationists and transportation planning professionals, which should be more interested in the practical results of the system and not in a detailed description of the algorithm. We have also clarified the novelty of the algorithms in the discussion in the third paragraph, lines 10-11: “Moreover, the algorithm of the MMS2 is much more robust than the previous version, which only used simpler image processing and computer vision methods and not machine learning.”

Point 2: In some cases values and figures are missing while in other cases, they are not interpreted clearly. I assume there are important results but when statements are not supported by findings it makes them pretty weak.

Authors: We have now added several figures to the manuscript. We also tried to clarify the interpretation of the results in the discussion.

Point 3: on the other hand - a statement like that in line 192 ('it can be trained to detect any animal group, objects or road features') is risky to be made if you have not tested that yourself or can rely on previous work.

Authors: The algorithm can be trained to detect any animal group or other objects, as it was able to detect some of the most difficult groups when road-killed (amphibians). We rephrased the sentence: “it could potentially be trained to detect any animal group, objects or road features”.

Point 4: What is the applicability of the system and technique to other parts off Evora? A short discussion is missing

Authors: The system is completely independent of the study area or species group. Indeed, as we stated previously, the MMS2 can be used to detect any object on the surface of the road: instead of detecting species it can be used to detect irregularities on the road. We included in the discussion the applicability of our system (fifth paragraph, line 2-3): “Our system can be used on national roads everywhere”.

Point 5: Throughout the manuscript, the authors give provide poor explanations concerning the results. They take it for granted that the reader would understand thus shortening descriptions. They should elaborate more (see the attached file for details)

Authors: We have now included more detailed explanations in the results and discussion.

Point 6: I am missing a Map of the area (Evora) including N4 and M529 and the rest of the area where the tests were implemented

Authors: We have now included in the manuscript the study area and surveyed roads (Figure 3).

Point 7: Table 1 contains the results of the controlled tests. Perhaps it would be beneficial to include also the results of the real detection in a summarized table as well

Authors: We decided to not present a table with the summarized results of the real surveys because we performed the tests with the same configurations, camera resolution, FPS and vehicle speeds. For this reason, we presented the results of the real surveys in text.

Point 8: English and grammar - should be improved. I have made several corrections but language editing is essential

Authors: We have checked carefully the spelling and grammar of the manuscript. Further, the manuscript has been corrected by an English native, Dr James Harris.

Response to the attached file

Point 9: Abstract – Line 16 – “…can be used by any person with or without sampling skills”: according to the disadvantages in discussion section - I doubt that

Authors: We mean that the sampling process in roads with the MMS2 could be performed by any person, even without sampling skills. Of course, the image processing and animal identification has to be performed by specialists. We edited the sentence to: “The MMS2 can be easily attached to the back of any vehicle and the surveys can be performed by any person with or without sampling skills” (abstract, lines 8-9).

Point 10: Abstract – Line 19

Authors: We replaced the word “missing” by “overlooking” as suggested by the referee.

Point 11: Introduction – Line 35

Authors: We replaced the part of the phrase “…makes the process somewhat dangerous for people as well, mainly in amphibian’s surveys, which are performed on rainy nights” by “makes the process somewhat dangerous for the surveyors, mainly during the survey of amphibians on rainy nights” as suggested by the referee (introduction, second paragraph, lines 4-5).

Point 12: Introduction – Line 36

Authors: We replaced the word “users” by “surveyors”.

Point 13: Introduction – Line 37

Authors: We replaced “not always the same” by “equal”.

Point 14: Introduction – Line 38

Authors: We replaced the word “missed” by “overlooked” throughout the manuscript.

Point 15: Introduction – Line 47

Authors: We replaced the word “by” by “of” and “was composed by” by “includes”.

Point 16: Introduction – Line 48: What is the a 'rough computer'? please explain

Authors: We replaced “rough computer” by “industrial computer”.

Point 17: Introduction – Line 49: on canvas or on map scale? not clear

Authors: On real scale. We replaced the phrase “…with a resolution of less than a millimetre” by “…obtaining an object resolution of less than one millimetre”.

Point 18: Introduction – Line 54-55: repetition of lines 33-39

Authors: We deleted the sentence as suggested by the referee.

Point 19: Introduction – Line 56

Authors: We replace the phrase “not very practical to use” by “cumbersome”

Point 20: Introduction – Line 56: of what? In compare to?

Authors: The MMS1 sampling width is not large in comparison to the road width. We completed the sentence: “…the road sampling width is not large (the camera only captures one metre of the road width, which is not enough to cover an entire one-way road)” (fourth paragraph, lines 10-11).

Point 21: Introduction – Line 57: duration?

Authors: The MMS1 was limited to the external source power, which lasted approximately 1 hour. We have now included this information in the introduction (fourth paragraph, line 12) and in Table 2.

Point 22: Introduction – Line 57-58

Authors: We replaced “the” by “an”, “overall” by “the total price of” and “something” by “somewhat”.

Point 23: Introduction – Line 59-60

Authors: We replaced the phrase “…mobile mapping system (MMS2) for detecting not only amphibians but small birds as well” by “…Mobile Mapping System (MMS2) for detecting amphibians and small birds” (fifth paragraph, lines 1-2).

Point 24: Introduction – Line 60: how considerable?

Authors: We have now included a figure illustrating the size of the system (Figure 1b).

Point 25: Introduction – Line 62: Why writing it here briefly when you elaborate in Materials and Methods?

Authors: We excluded the sentence as suggested.

Point 26: Materials and methods – Line 70

Authors: We replaced “it is composed by” by “it includes”.

Point 27: Materials and methods – Line 77: Not clear. How compact and what do you mean by autonomy? Please explain

Authors: We mean that the high-power processing computer is as compact as a laptop. By unlimited autonomy we mean that the system does not depend on the laptop battery for instance. It can work as long as the vehicle is working due to the lighter charger. We excluded the sentence “It is compact and can be connected to the vehicle charger for unlimited autonomy”, but included this information at the point 5) of the same paragraph of the Materials and methods: “Lighter charger for vehicles, for unlimited sampling length (does not depend on an external power source, but rather depend on the car battery; the system runs uninterruptedly as long as the vehicle is working).”.

Point 28: Materials and methods – Line 79: What is the LBS stands for?

Authors: LBS stands for Location-Based Service. We included this information in the manuscript (materials and methods, first paragraph, point 3).

Point 29: Materials and methods – Line 85: If the authors thought it is important to elaborate on the development languages, then they should explain the combination of C++, Java, python and PHP for the development of desktop application. Besides, PHP is a server-side language and needs a web-server in order to be implemented. Please clarify

Authors: The PHP code was included in order to be possible to remotely send the captured images, but it is not currently being used. We decided to not explain the combination of development languages used, so we excluded the languages from the manuscript.

Point 30: Materials and methods – Line 85

Authors: We replaced the phrase “and a software that inserts the device coordinates continuously along the way” by “and a software that pin-points continuously the device coordinates along the way” as suggested by the referee (methods, description of the MMS2, second paragraph, lines 3-4).

Point 31: Materials and methods – Line 92: How large?

Authors: The higher the number of images, the better the algorithm will perform. To our knowledge, there is no fixed rule for how many images are necessary. Although, there is a threshold in which the algorithm will start to perform poorly (overfitting). In order to know this value, we would need a bigger image database and much more tests.

Point 32: Materials and methods – Line 94: Are there any thresholds?

Authors: As stated previously, there is a limit number of images in which the algorithm will start overfitting, but we do not know this number yet as much more tests would be necessary.

Point 33: Materials and methods – Line 95

Authors: We kept the expression “We performed” as we think it is more correct in this particular phrase.

Point 34: Materials and methods – Line 97

Authors: We replaced “We selected” by “We thus selected”.

Point 35: Materials and methods – Line 97: How? To what extent? Based on what indicators?

Authors: A simple test was done using 150 images from which 25 contained animals. Then, we tested different architectures from the CNN, and measured the efficiency of these architectures to detect the animals (VGG19: 42%, ResNet50: 35%, Inception V3: 62%, 97 Xception: 23%, VGG16: 62%). This was done with the Keras library pre-trained models, using a set of 600 images to post train the algorithm. The VGG16 presented better results so we decided to follow its path. At this stage, the false positives were not considered as they could be improved later. We included this information in the methods (algorithms, second paragraph, lines 3-6).

Point 36: Materials and methods – Line 101: Please explain/elaborate on POOL and CONV. Not clear

Authors: Pooling is basically “downscaling” the image obtained from the previous layers, to reduce its pixel density. The most popular kind of pooling used is Max Pooling. If we wanted to pool by a ratio of 2 (meaning that the image height and width will be half of the original size), we would need to compress every 4 pixels (a 2x2 grid) and map it to a new single pixel without loss of “important” data from the missing pixels. Max Pooling is done by taking the largest value of those 4 pixels. Thus, one new pixel represents 4 old pixels by using the largest value of those 4 pixels. This is done for every group of 4 pixels throughout the whole image. The convolutional layer is the main building block of a convolutional network. It is the layer that will compute the output of neurons that are connected to local regions in the input, each computing a dot product between their weights and a small region they are connected to the input volume. We have now tried to explain more clearly these concepts in the manuscript (methods, algorithms, fourth paragraph).

Point 37: Materials and methods – Line 110: Not clear, please elaborate and explain: accuracy of what; generalization of what?

Authors: The referred processes reduce overfitting because they reduce the number of times the algorithm will make a false positive; and improve the accuracy of animal detection means because they increase the rate of correct classifications, not creating as much false positives as well. We included this information in the manuscript (methods, algorithms, fourth paragraph, lines 15-17).

Point 38: Materials and methods – Line 115: What tests? How?

Authors: We performed some preliminary tests to adjust the camera configurations. We just tried different configurations and checked the images captured (whether they were under or over-exposed, or whether they provided the real colours, etc.). We carried out these tests in the surveyed roads. We added this information in the manuscript (methods, testing framework, first paragraph): “For that, we went through the surveyed roads in different days with different light and weather conditions and verified the captured images, changing the camera configurations every time (exposure, white balance, brightness, contrast, hue and saturation) until finding the optimal combination of parameters.”

Point 39: Materials and methods – Line 124: verges of what/where?

Authors: In the road verges. We have now completed the phrase (second paragraph, line 7).

Point 40: Materials and methods – Line 131: The site is in Portuguese.

Authors: We replaced the Portuguese link with the English link (https://lifelines.uevora.pt/?lang=en).

Point 41: Materials and methods – Line 136: Since you have declared at the beckoning that the MMS2 is less time-consuming then 'regular' survey, I would have expected to see comparisons and figures between the tests to emphasize what is stated at the introduction

Authors: In this manuscript, the objective was not to compare the performance between regular surveys and the MMS2. For that, we would need more tests, so we decided to leave the comparison for later when we have enough data. In this paper, the main objective was to test the effectiveness of the MMS2 in detecting road-killed animals.

Point 42: Results – Line 140-141

Authors: We excluded “We found…” and included only “The best configuration…” and we replaced the word “compromising” with “in relation to” as suggested by the referee.

Point 43: Results – Line 143

Authors: We replaced “at the direction of the road, which results…” by “of a given road, resulting…”.

Point 44: Results – Line 146: Rephrase. Not clear.

Authors: We replaced the phrase “…an animal detection rate between 63.3 and 80% and between 17.4 and 24% of false positives” with “…a rate of correct classification (i.e. a percentage of detected animals) between 63.3 and 80% and a percentage of false positives (i.e. percentage of frames in which the algorithm detected a road-killed animal incorrectly) between 14.4 and 24%” (results, second paragraph, lines 1-4).

Point 45: Results – Line 148

Authors: We replaced “that performed the best” by “with best performance”.

Point 46: Results – Line 151: What does the 'mean detection rate' stands for? This is not what you have written in the abstract. The terms 'real', 'roads', 'surveyed roads', 'detected animals', 'detected images' have to be consistently explained. At the moment it is a collection of numbers and figures that make little sense to the reader.  

Authors: We considered the mean detection rate as the mean percentage of correct positives from the total number of frames with animals. We replaced the term “mean detection rate” by “rate of correct classification” throughout the manuscript. We have now tried to be more consistent with the terminology used and we explained each term used in the results (second paragraph and Table 1).

Point 47: Results – Line 154: How much for the tests and how much for the real?

Authors: The real surveys originated approximately 1.100.000 images and the controlled surveys approximately 10.000 images.

Point 48: Discussion – Line 160-168: You already said that but did not discuss why the improved algorithm overlooked 22% when the previous version overlooked only 16%.

Authors: The MMS1 failed to detect only 16% of the animals on roads, while the MMS2 failed 22%. The reason for this might be the absence of lighting system in the MMS2, which results in parts of the road being not equally illuminated, and the algorithm cannot detect the animal in the darkest areas. Moreover, the MMS1 had a better image resolution and lower image distortion. We have now included this in the discussion (second paragraph, lines 7-9).

Point 49: Discussion – Line 175: Here you write that a difference of 3% is a big advantage while previously you refer to an error of 6% (22-16) as an acceptable error. Please to explain why 

Authors: It is not a difference of 3%. The MMS1 originated 80% of false positives, while the MMS2 originated 17% of false positives. We have now reorganized the sentence in order to make it clearer (discussion, third paragraph, lines 2-4).

Point 50: Discussion – Line 175

Authors: We replaced “manually verifying” with “manual verification of”.

Point 51: Discussion – Line 176: How consuming? To what extent?

Authors: The real tests of the MMS1 originated 9641 images in a total distance of 30 km, and the algorithm detected 232 images, 47 with amphibians and 185 without (Sillero et al. 2018). The road-kill monitoring program of University of Évora survey for around 100 km each day. Therefore, the MMS1 would originate around 30.000 images for each surveyed day. In fact, the MMS1 generated a unique image with the total length of the survey: if the survey was 30 km, the image was 30 km long as well. For an easier processing, the MMS1 images were split in images of 1 m. If the number of false positives was proportional, it would create daily more than 500 false positives. Manually verifying more than 500 images every day would take several hours.

Point 52: Discussion – Line 177-182: 2-4 are merely technological improvements. They are important but emphasise reduces the novelty of the algorithm. As I understand, this is the core of the study and it is left aside in the discussion.

Authors: We excluded the points 2-4 from the discussion. We have now included the novelty of the algorithm in the discussion (third paragraph, lines 8-11).

Point 53: Discussion – Line 181: Any road width?

Authors: We meant a one-way national road width, that are usually 6-7 meters. We included this information in Table 2.

Point 54: Discussion – Line 183: So, it is not easy to use by 'any person with or without sampling skills'. If a human intervention is required, it is not easy to use by anyone.

Authors: As stated before, we mean that the system can be used in the field for surveying by any person, but the image processing and identification is made by specialists. We added this information in the discussion (fourth paragraph, line 7-8).

Point 55: Discussion – Line 185: Not clear

Authors: In particular weather conditions, the images obtained may not be of enough quality to identify the animals. We rephrased the sentence: “the weather conditions may affect the images’ quality, hampering the correct animal identification” (fourth paragraph, lines 2-3).

Point 56: Discussion – Line 192: But it wasn’t tested, so how can you be so sure?

Authors: As we stated before, the algorithm can be trained to detect any animal group or other objects, as it was able to detect some of the most difficult groups when road-killed (amphibians). We rephrased the sentence: “it could potentially be trained to detect any animal group, objects or road features” (discussion, fourth paragraph, line 11).

Point 57: Table 1 – Line 261: Please provide in the table caption the main 1-2 insights that can be observed.

Authors: We added the phrase to the table caption: “The best configuration resulted in a correct classification of 80% and originated 20% of false positives”.

Point 58: Table 1 – Line 268: Out of what? Please explain

Authors: It is now explained better in the caption of Table 1: “False positives: number of frames the algorithm detected a road-killed animal incorrectly”.

Round 2

Reviewer 1 Report

Thanks to the authors for the corrections. I have no further reservations. Minor editing would do better.  It was just about some editing the language. 

Author Response

We appreciate the referees’ comment and we have now carefully reviewed the language, with minor editions highlighted in yellow.

Reviewer 2 Report

After the review, the paper has improved a lot. The work is described in detail and shows an interesting use of image recognition and ML, and an excellent work of implementation.

Author Response

We appreciate the referees’ comment and also the previous useful suggestions for improving the manuscript.

Reviewer 3 Report

I have read the revised manuscript and the response of the authors.  The manuscript has been significantly approved and is now, in my opinion, suited for publication.

Author Response

Thank you very much for your comment and for reading our revised manuscript and response letter. We have now revised the language and introduced some minor corrections, highlighted in yellow.